



# Hydrometeor partitioning ratios for dual-frequency space-borne and polarimetric ground-based radar observations

Velibor Pejcic[1], Kamil Mroz[2], Kai Mühlbauer[1], and Silke Trömel[1,3]

[1]Institute of Geosciences, Department of Meteorology, University of Bonn, Bonn, Germany
[2]National Center for Earth Observation, University of Leicester, Leicester, UK
[3]Laboratory for Clouds and Precipitation Exploration, Geoverbund ABC/J, Bonn, Germany

**Correspondence:** Velibor Pejcic (velibor@uni-bonn.de)

**Abstract.** Conventional radar-based hydrometeor classification (HMC) algorithms identify the dominant hydrometeor type within a resolved radar volume, while more recent techniques allow estimation of the proportions of individual hydrometeor classes (hydrometeor partitioning ratios, HPRs) within a mixture. These newer algorithms ($\mathrm{HMC_P^{DP}}$) are based on dual-polarization (DP) measurements from ground-based radars (GR), while similar algorithms do not yet exist for space-borne radars (SR) with dual-frequency (DF) capabilities. This study has three objectives, (1) to evaluate HPR retrievals, (2) to exploit the combination of DF SR and DP GR for estimating HPR based on satellite DF observations ($\mathrm{HPR_k^{DF}}$) and (3) to further improve $\mathrm{HPR_k^{DP}}$ estimates based on GR DP observations. To achieve these, DP measurements of NEXRAD's GRs are matched with those of the dual-frequency precipitation radar of the Global Precipitation Measurement Core satellite. All matched volumes are represented by averaged DF and DP observations and several hundred GR sub-volumes classified with the standard HMC. The latter are used to calculate quasi-HPRs (qHPRs). qHPRs and averaged DF and DP variables serve as basis for the $\mathrm{HPR_k^{DF}}$ and $\mathrm{HPR_k^{DP}}$ retrievals, which in turn are evaluated with the qHPRs. The vertical distributions of $\mathrm{HPR_k^{DF}}$ and $\mathrm{HMC_P^{DP}}$ products are in good agreement. Furthermore, the estimated HPRs show for most hydrometeor classes high correlations with the qHPRs and confirm the overall good performance of the algorithms. However, $\mathrm{HMC_P^{DP}}$ performance is superior to $\mathrm{HMC_P^{DF}}$. In both DF and DP space, snow HPRs are underestimated, graupel HPRs are overestimated, and HPRs for big drops show only low correlations.

## 1 Introduction

Hydrometeor classifications (HMC) using ground-based polarimetric weather radars (GR) observations play an essential role, e.g. to refine quantitative precipitation estimation (Giangrande and Ryzhkov, 2008), to detect hail and estimate its size and damage potential (Ortega et al., 2016; Ryzhkov et al., 2013; Ackermann et al., 2023) and to identify freezing rain (Thompson et al., 2014), which can serve as a warning system for transport infrastructure (Trömel et al., 2017). The majority of HMCs identify the dominant hydrometeor type within each resolved radar volume exploiting measurements from dual-polarization (DP) weather radars and specific classification methods. The most commonly used classification methods are based on the fuzzy logic approach (Dolan and Rutledge, 2009; Dolan et al., 2013; Zrnić et al., 2001a; Straka et al., 2000; Thompson et al., 2014; Ribaud et al., 2016; Park et al., 2009), but there are also methods that rely on the Bayesian approach (Yang et al., 2019;



Marzano et al., 2007) or clustering techniques (Grazioli et al., 2015; Ribaud et al., 2019; Lukach et al., 2020; Besic et al., 2016). More detailed description of hydrometeor mixtures (Besic et al., 2018) are the so-called hydrometeor partitioning ratios (HPRs), which represent estimates of the proportion of the polarimetric signal originating from a specific hydrometeor class within a resolved radar volume. Besic et al. (2018) provided a methodology to estimate HPRs, which was subsequently refined in Trömel et al. (2023). HPRs have recently been utilized to study microphysics and dynamics of precipitation (Gehring et al.,

2020, 2022), to verify microphysical retrievals (Billault-Roux et al., 2023; Planat et al., 2021) and to evaluate hydrometeor distributions in NWP models (Trömel et al., 2021; Vignon et al., 2019; Jang et al., 2021; Shrestha et al., 2022; Trömel et al., 2023).

Only a few space-borne measurement platforms with radars exist or have existed in the past: CloudSat (Stephens et al., 2002), designed for observations of clouds and light precipitation, the Tropical Rainfall Measuring Mission (TRMM; Liu

et al., 2012), which is the first precipitation satellite with a $K_u$-band precipitation radar (PR) on board, and its successor the Global Precipitation Measurement core satellite (GPM) with the first Dual-Frequency Precipitation Radar (DPR) measuring precipitation at $K_u$-band and $K_a$-band frequencies (Hou et al., 2014). Rain rates estimated from space-borne radars (SR) are significantly affected by the hydrometeor types located within a resolved measurement volume (Liao and Meneghini, 2022). SR-derived HMCs, using the DPR e.g., are based on very simple subdivisions of the hydrometeors. The detection of the melting

layer (ML) top and bottom is used to distinguish between solid, liquid and melting hydrometeors (Le et al., 2016). Additional two-dimensional classifications are provided for snow (*flagSurfaceSnowfall;* Le et al., 2017)), graupel/hail (*flagGraupelHail;* Le and Chandrasekar, 2021a) and hail (*flagHail;* Le and Chandrasekar, 2021b) and are based on the so-called precipitation type index (PTI). The PTI is derived from the storm top height (STH), the maximum measured reflectivity at $K_u$-band and the average slope of the dual-frequency ratio profile. Mroz et al. (2017) presented several hail detection algorithms based on

DF profile observations but also on brightness temperature measurements of GPMs Microwave Imager (GMI). All products do not provide information on the vertical distribution of these hydrometeor classes and are not considered in DPRs rain rates estimation (Iguchi et al., 2010). Seiki (2021) was the first to develop a three-dimensional HMC based on dual-frequency (DF) measurements, but only for hail detection.

In this study, the HMC scheme from Trömel et al. (2023) ($HMC_P$), estimating HPRs in DP-space ($HMC_P^{DP}$), is refined and

extended to the DF-space ($HMC_P^{DF}$). For this purpose, satellite-based DF observations from GPM's DPR are combined with ground-based DP measurements from NEXRAD's S-band WSR-88D radars. In order to combine the high-resolution GR and the low-resolution SR data, the DF and DP measurements are averaged to obtain data with approximately equal volumes, so-called superobbed data. Each superobbed observation then contains information about the partitioning ratios of the different dominant hydrometeor classes (quasi hydrometeor partitioning ratio, qHPR), approximated by the relative occurrences of the

dominant hydrometeor classes in high-resolution radar bins within the supperobbed volume as determined by conventional DP-based HMC. These qHPRs are used as a basis for the derivation of the HPRs in DF and DP space. Subsequently, the HPRs estimated with $HMC_P$ from either superobbed DF or DP measurements are validated using the qHPR estimates.



Sect. 2 introduces the SR and GR measurements and their processing, followed by the explanations of the merging procedure and the qHPR derivation. Sect. 3 explains the methodology for HPR estimates. The results are shown in Sect. 4 followed by a conclusion in Sect. 5.

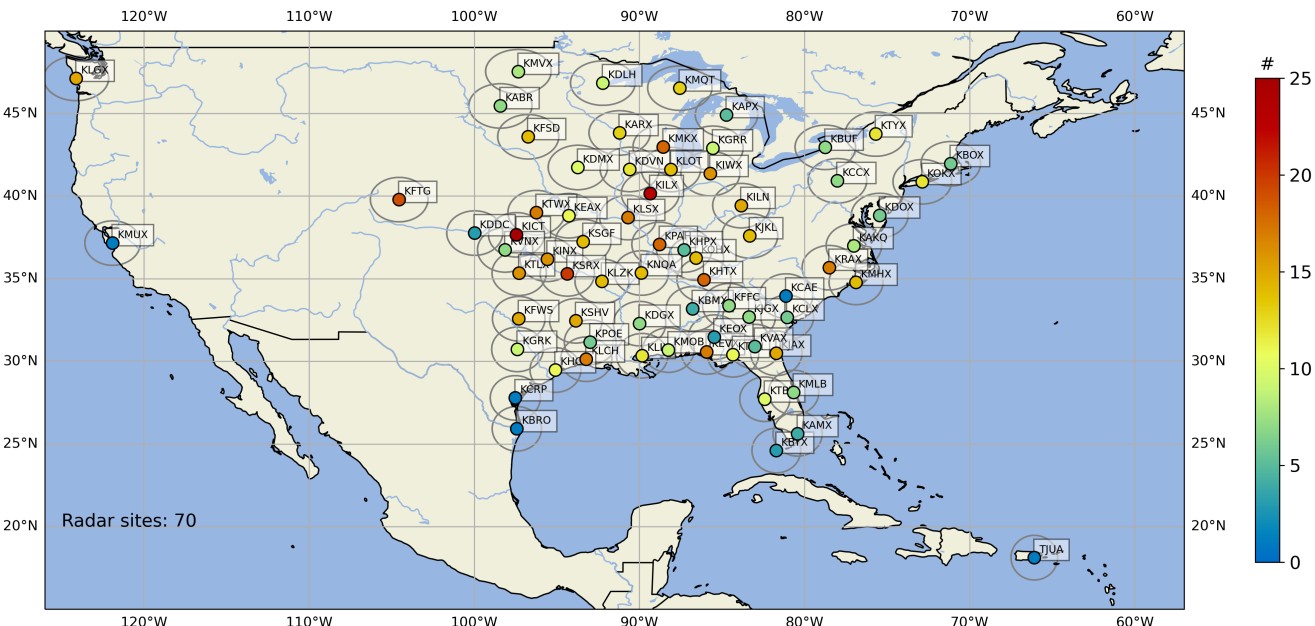

**Figure 1.** Overview of the NEXRAD weather radar (WSR-88D) sites used in this study. The 150 km range for an elevation angle of 0.5° is illustrated as gray circle. The colored dots indicate the location of the respective radar and the number of GPM overpasses in the period between 2014 and 2023 used in this study. The total number of used radar sites is indicated in the lower left corner.

## 2 Data

### 2.1 Space-borne radar observations

The DPR onboard the GPM Core Observatory (Iguchi and Meneghini, 2021) comprises two radars: the $K_u$-band Precipitation Radar ($K_uPR$, 13.6 GHz) and the $K_a$-band Precipitation Radar ($K_aPR$, 35.5 GHz). The DPR provides measurements with a vertical resolution of 250 m, over-sampled every 125 m, and a horizontal resolution of approximately 5 km due to the satellite's altitude of 407 km before the orbit boost in November 2024 (Kubota et al., 2024).



Initially, the $K_u$PR operated across a 245 km wide swath (49 beams), while the $K_a$PR was limited to a narrower central swath of 125 km (25 beams), nested within the $K_u$PR swath. The $K_a$PR employed two distinct scanning modes: Measurements with a vertical resolution of 250 m, fully overlapping the central part of the $K_u$PR swath (High-Resolution Mode).

Measurements with a vertical resolution of 500 m, where the scan pattern was laterally shifted by half a footprint (24 beams) in the cross-track direction (Shifted Scan Mode, Hou et al., 2014).

On 21 May 2018, the scanning strategy was updated to extend the $K_a$PR swath to 245 km, matching the $K_u$PR swath width. The 24 beams were moved to the outer parts of the swath. This adjustment ensured that all footprints in the extended $K_a$PR swath included DF measurements, significantly enhancing data consistency and coverage (Iguchi et al., 2010). Single-

frequency beams are not considered in this study. To derive parameters of the drop size distribution (DSD), precipitation rates and attenuation corrected $K_u$-band and $K_a$-band reflectivities in logarithmic space, the measured $K_a$-band ($Z_{K_a}^m$) and $K_u$-band ($Z_{K_u}^m$) reflectivities are processed in various modules described in more detail in Iguchi et al. (2010). The DF ratio

$$DFR_{K_u-K_a}^m = Z_{K_u}^m - Z_{K_a}^m \tag{1}$$

is the difference between $Z_{K_u}^m$ and $Z_{K_a}^m$ reflectivities in logarithmic space. In stratiform precipitation $DFR_{K_u-K_a}^m$ is mainly

affected by non-Rayleigh scattering effects and path-integrated attenuation. In the solid phase, attenuation by frozen hydrometeors is negligible for both frequencies and does not significantly change $DFR_{K_u-K_a}^m$. In contrast, the non-Rayleigh scattering effects play a major role and lead to an increase of the $DFR_{K_u-K_a}^m$ with increasing hydrometeor diameters (Le et al., 2016; Iguchi et al., 2018). Also, $DFR_{K_u-K_a}^m$ in the solid region depends on the density of hydrometeors and their degree of riming. According to the Mie theory an increase in $DFR_{K_u-K_a}^m$ is expected with decreasing density of fluffy non-rimed solid hydrom-

eteors with low $Z_{K_u}^m$ (Seiki, 2021). For a fixed $DFR_{K_u-K_a}^m$ $Z_{K_u}^m$ increases with the ice particles degree of riming. However, this is only valid in stratiform precipitation and if $DFR_{K_u-K_a}^m > 1$ dB (Tridon et al., 2019). In the melting layer (ML) we observe an increase in $Z_{K_u}^m$ due to the changes in the refractive index, particle size and concentration (Ryzhkov and Zrnic, 2019). As a consequence, both non-Rayleigh scattering effects and attenuation increase the $DFR_{K_u-K_a}^m$ and result in a pronounced "bump", called the $DFR_{K_u-K_a}^m$ bright band, in the vertical profile of the $DFR_{K_u-K_a}^m$ (Le et al., 2016). In the liquid phase, attenuation

mainly controls the $DFR_{K_u-K_a}^m$. $K_a$-band measurements are much more affected by attenuation compared to measurements at $K_u$-band and lead to an increase in $DFR_{K_u-K_a}^m$ towards the ground. This increase is even more pronounced in convective precipitation where higher precipitation rates, thus the attenuation values, are observed. Furthermore, convection promotes the presence of large hydrometeors such as graupel, hail or drop diameter exceeding 0.8 mm (Mroz et al., 2024) that contribute to non-Rayleigh related $DFR_{K_u-K_a}^m$ increase. In deep convective cores, the typical vertical profile of $DFR_{K_u-K_a}^m$ can be dis-

torted by multiple scattering at $K_a$-band. In extreme MS conditions, attenuation of the high frequency radar observations is compensated by multiple scattering effects in the upper part of the atmosphere which results in the so-called $DFR_{K_u-K_a}^m$-knee, i.e. a decrease in $DFR_{K_u-K_a}^m$ towards the ground (Battaglia et al., 2014).

The overall vertical structure of the $DFR_{K_u-K_a}^m$ is used to categorize the measurements into different rain types (stratiform, convective, other, see Le et al. (2016)) and to determine the ML thickness and height, and is used to distinguish between liquid,

solid and melting precipitation regions (Iguchi et al., 2010; Le and Chandrasekar, 2012).





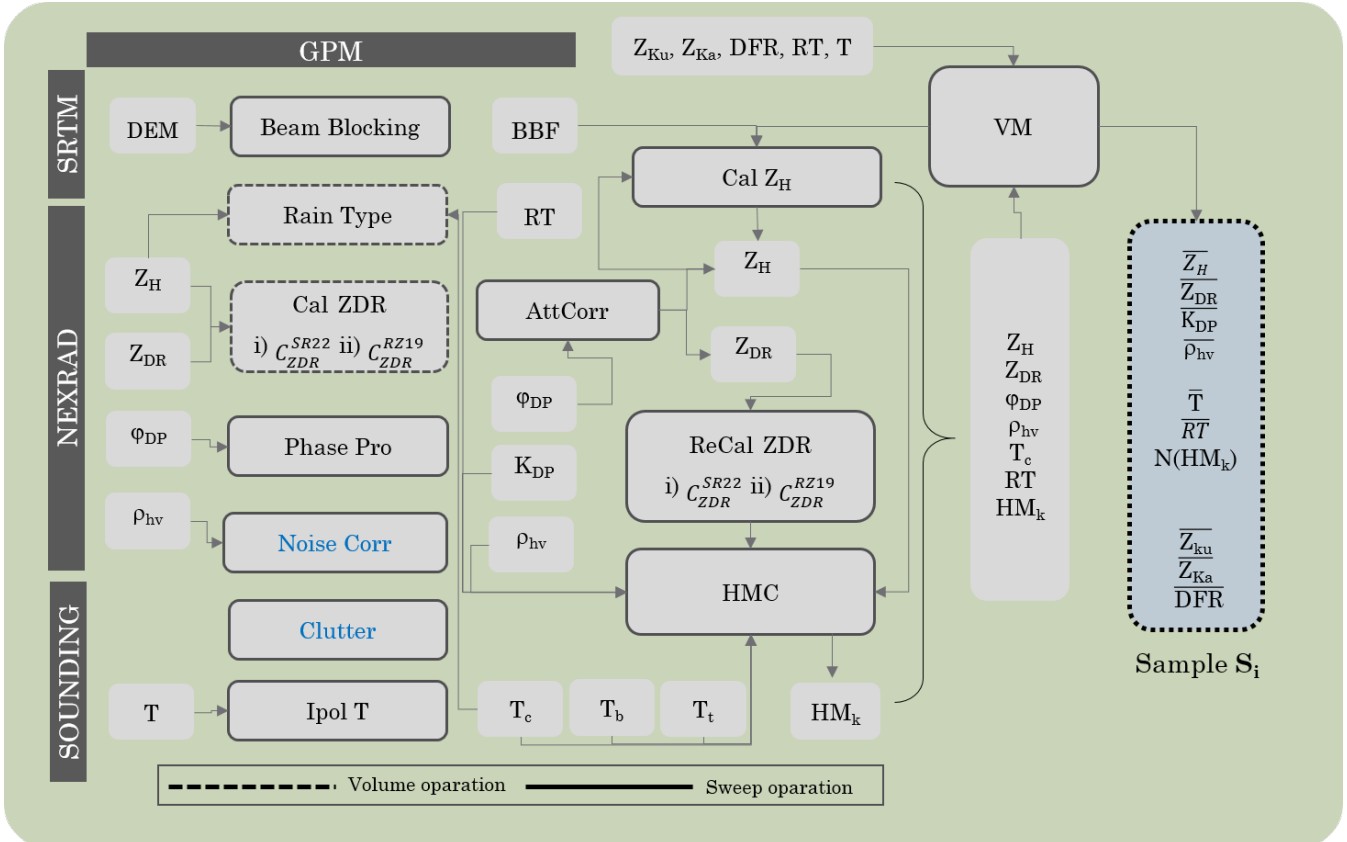

**Figure 2.** Workflow for polarimetric radar data processing of the NEXRAD S-band weather radars (gray boxes). The black boxes represent the different data sources used and the gray boxes outlined with solid or dashed lines represent processing operations based on a sweep or volume data, respectively. Operations that have already been performed on the NEXRAD data are written in blue.

## 2.2 Ground-based radar observations

DP measurements of the NEXRAD WSR-88D S-band weather radars are exploited for this study. In total, 757 volume scans measured between 2014 and 2023 of the radar sites shown in Fig. 1 are considered. Measurements were selected to ensure that the GPM overflight took place at the closest point in time. A balanced number of convective and stratiform events is maintained to ensure a good representation of less frequently occurring hydrometeors like hail. The range resolution of the utilized NEXRAD radars is 250 m with a maximum elevation angle of $19.5°$ and $1°$ degree azimuthal resolution for higher elevation. Quality-controlled GR observations are provided by NASA's GPM Ground Validation program (GPM-GV). The quality control includes non-precipitating echos removal with different thresholds and phase unfolding. In addition, GPM-GV also provides vertical temperature information from model soundings (Pippitt et al., 2013). Additional GR processing (Fig. 2) is explained in more detail in the following:



The vertical temperature profiles are interpolated linearly at the beam center ($T_c$) and at the respective outer beam edges (3 dB beam width). From now on referred to $T_t$ as temperature at the top beam edge and $T_b$ as the temperature at the bottom beam edge. All radar bins with $T_b<0$°C are classified as solid and all those with $T_t>4$°C as liquid. All other radar measurements are considered as partly melted. Digital Elevation Model (DEM) data from the Shuttle Radar Topography Mission (SRTM, Reuter

et al. (2007)) is used to calculate any possible beam blocking fractions (BBF) following Bech et al. (2003). $Z_H$ is smoothed with a moving average of 5 range bins, while 11 range bins are used to smooth $Z_{DR}$ and $\rho_{HV}$. A $\rho_{HV}$ threshold of 0.8 is applied for the noise filtering. In the next step, the rain type classification following Park et al. (2009) is applied to the entire volume to classify convective and stratiform radar bins, but with slight modifications (i.e., the classification as convective based on $\rho_{HV}$ only is omitted).

For $Z_{DR}$ calibration, either the method using Quasi-Vertical-Profiles (Sanchez-Rivas and Rico-Ramirez, 2022) in the following referred to $\mathrm{Cal}_{Z_{DR}}^{SR22}$ or the $Z_H$-$Z_{DR}$ consistency in light rain (Ryzhkov and Zrnic, 2019), referred as $\mathrm{Cal}_{Z_{DR}}^{RZ19}$, is applied. Since the data base is limited to volume scans for specific time steps only, $\mathrm{Cal}_{Z_{DR}}^{SR22}$ is not applied to Quasi-Vertical-Profiles but to all available PPI scans to include a larger amount of data in the calibration routine. Slight modifications of $\mathrm{Cal}_{Z_{DR}}^{SR22}$ include the application of the median instead of the mean (Eq. 10; Sanchez-Rivas and Rico-Ramirez, 2022) for noise filtering

and recalculation of the intrinsic mean $Z_{DR}$ (0.178 dB) for the S-band data. A first guess $Z_{DR}$-offset, using either $\mathrm{Cal}_{Z_{DR}}^{SR22}$ or $\mathrm{Cal}_{Z_{DR}}^{RZ19}$ if there are less then 1000 valid radar bins, is applied on the entire volume scan before the final recalculated $Z_{DR}$-offset is applied after correction for (differential) attenuation sweep-wise. Valid observations for the $Z_{DR}$-offset calibration are all radar bins with $\rho_{HV}> 0.99$, $T_t>5$°C and if applying $\mathrm{Cal}_{Z_{DR}}^{SR22}$ $0\,\mathrm{dBZ}\leq Z_H \leq 20\,\mathrm{dBZ}$ otherwise $20\,\mathrm{dBZ}\leq Z_H \leq 30\,\mathrm{dBZ}$ for $\mathrm{Cal}_{Z_{DR}}^{RZ19}$.

The processing of differential Phase $\phi_{DP}$ includes radial smoothing with a window size of 9 radar bins for measurements $Z_H >40\,\mathrm{dBZ}$ (heavy rain) and a window size of 25 radar bins elsewhere (Park et al., 2009). Instead of determining $K_{DP}$ based on the slope of a least squares fit, a low-noise Lanczos differentiator (Heistermann et al., 2013; Diekema and Koornwinder, 2012) is used to speed up the processing significantly. The two window sizes are also applied for the $K_{DP}$ derivation. Correction for (differential) attenuation applies parameters $\alpha = 0.04\,\mathrm{dB\,deg^{-1}}$ and $\beta = 0.004\,\mathrm{dB\,deg^{-1}}$ (Ryzhkov and Zrnic, 1995). The

attenuation correction is limited to the liquid phase ($T_t > 4$°C) and the highest values of the path-integrated attenuation (PIA) and path-integrated differential attenuation (PIDA) reached in the liquid phase are applied to the remaining mixed phase and solid radar observations.

$Z_H$ calibration ($\mathrm{Cal}_{ZH}$) is performed by comparing the GR with SR measurements (Pejcic et al., 2022; Crisologo and Heistermann, 2020; Warren et al., 2018; Louf and Protat, 2023; Protat et al., 2022). GR and SR measurements are matched to

the same geometry for each volume scan (more detailed description in Sect. 2.3), but measurements contaminated by the ML are excluded from the offset calculations (Pejcic et al., 2022). For this purpose, the ML top and bottom estimates determined by the DPR are used. The conversion of reflectivity from $K_u$-band to S-band wavelengths is performed following Cao et al. (2013). Further refinements of $\mathrm{Cal}_{ZH}$ include the use of quality indices, determined from BBF and PIA, as weighting factors for determining the $Z_H$-offset (Crisologo and Heistermann, 2020).



The applied standard HMC ($\mathrm{HMC_Z}$ Zrnić et al., 2001b) to identify the dominant hydrometeor type in a resolved radar volume and used to estimate the qHPRs is based on two dimensional membership functions (MSF) defined in Park et al. (2009) with slightly modified hydrometeor types and MSF-parameters. The predefined hydrometeor types are light rain (LR), moderate rain (MR), heavy rain (HR), big drops (BD), rain/hail (RH), graupel (GP), crystals (IC), dry snow (SN), wet snow (WS), plates/dendrites (PD) and hail (HA). The hydrometeor classes are generally abbreviated to $\mathrm{HM_k}$ where $k = 1, \ldots, n$ with n=11. For more information on $\mathrm{HMC_Z}$, we refer to the appendix A .

## 2.3 GR-SR Merging

The volume matching method (VMM) is performed with $\omega$radlib (Heistermann et al., 2013) and represents a well-known method for transferring SR and GR measurements to comparably sized volumes. In a first step all DP measurements of all GR bins within the SR footprint are averaged ($\overline{\mathrm{DP}}$, Fig. 3, left, plan view). Secondly, the DF observations of all SR bins (vertical resolution 125 m) within the GR beamwidth are averaged ($\overline{\mathrm{DF}}$, Fig. 3, right, side view). For more details see Warren et al. (2018) or Pejcic et al. (2022). This results in equally sized superobbed volumes described by averaged DP variables $\overline{Z}_\mathrm{H}$, $\overline{Z}_\mathrm{DR}$, $\overline{K}_\mathrm{DP}$ and $\overline{\rho}_\mathrm{HV}$ and averaged DF variables $\overline{Z}^\mathrm{m}_\mathrm{K_u}$, $\overline{Z}^\mathrm{m}_\mathrm{K_a}$ and $\overline{\mathrm{DFR}}^\mathrm{m}_\mathrm{K_u-K_a}$, from now on called sample $\mathrm{S_i}$ (Fig. 3, top center and Fig. 2, blue box). Furthermore, each $\mathrm{S_i}$ contains a mean temperature ($\overline{\mathrm{T}}$) and a rain type index ($\overline{\mathrm{RT}}$). $\overline{\mathrm{RT}}$ is convective if more then 10% of the GR pixels in a $\mathrm{S_i}$ are defined as convective ($\overline{\mathrm{RT}} = 2$), otherwise $\overline{\mathrm{RT}}$ is defined as stratiform ($\overline{\mathrm{RT}} = 1$). $\mathrm{S_i}$ also includes the number of dominant hydrometeor classes $\mathrm{N(HM_k)}$ classified with $\mathrm{HMC_Z}$ on the original GR radar grid (Fig. 2, blue box). For each $\mathrm{S_i}$ the $\mathrm{N(HM_k)}$ are used to calculate the $\mathrm{qHPR_k}$ via

$$qHPR_k = \frac{N(HM_k)}{\sum_{k=0}^n N(HM_k)}. \tag{2}$$

Note that qHPRs only represent estimators for the HPRs. E.g., due to their disproportionate influence on the polarimetric moments, hail or graupel may be classified as the dominant hydrometeor class in radar volumes despite low HPR. This can lead to overestimated qHPR for graupel and hail. In this study only $\mathrm{S_i}$ with at least 50 valid GR pixels, well-defined stratiform or convective SR profiles and DPR detected precipitation (*flagPrecip*) are considered. $\mathrm{S_i}$ showing strong differential attenuation due to hot spots above the ML or depolarization streaks (Ryzhkov and Zrnic, 2019) leading to negative $Z_\mathrm{DR}$ stripes are excluded. Furthermore, SR observations below 15.5 dBZ at $\mathrm{K_u}$-band (Liao and Meneghini, 2022) and 18 dBZ at $\mathrm{K_a}$-band (Mroz et al., 2024) are not considered.

80% of the $\mathrm{S_i}$, including $\overline{\mathrm{DF}}$, $\overline{\mathrm{DP}}$ measurements and qHPRs, serve as training data for the $\mathrm{HMC_P}$ (Fig. 3, center) and the remaining 20% of the $\mathrm{S_i}$ are utilized as training dataset for the evaluation. Sect. 4.1 presents results for one case study entirely independent of the test and training dataset used.





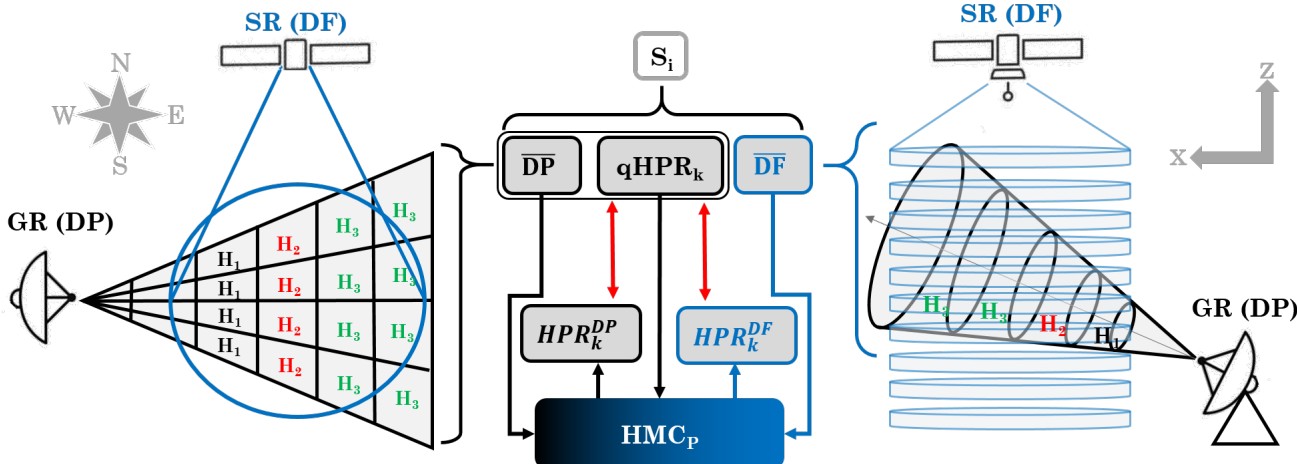

**Figure 3.** Schematic illustration of the workflow to derive and evaluate the HPR with $HMC_P$ based on $\overline{DF}$, $\overline{DP}$ and $qHPR_k$ (center) by comparing SR (in blue) and GR (in black) observations. The plan view on the left and the side view on the right site. The hydrometeor classes are indicated with colored $H_1$, $H_2$ and $H_3$

## 3 Hydrometeor partitioning Ratios (HPR)

In the following, we interpret the polarimetric measurements as multidimensional vectors and thus assume HPRs can be de-
termined based on multidimensional distribution functions $p_k$ for the different hydrometeor classes. If the multidimensional measurement approaches the mean of a specific $p_k$, the $HPR_k$ for that particular hydrometeor class k, increases, and vice versa, the farther away it is, the smaller the $HPR_k$ becomes (Besic et al., 2018; Trömel et al., 2023). Trömel et al. (2023) introduced $HMC_P$ as a modified version compared to Besic et al. (2018). This section details further advancements of $HMC_P$ and additionally transfers the methodology from DP to DF observation space.

The supperobbed variables in samples $S_i$ are stored in the multidimensional observation vector

$$\boldsymbol{X_i}^{DP} = \begin{bmatrix} \overline{Z}_H \\ \overline{Z}_{DR} \\ \overline{K}_{DP} \\ \overline{\rho}_{HV} \\ \overline{RT} \end{bmatrix} \tag{3}$$

including the averaged DP variables $\overline{Z}_H$, $\overline{Z}_{DR}$, $\overline{K}_{DP}$ and $\overline{\rho}_{HV}$, together with the rain type index $\overline{RT}$. Similarly, the multidimensional DF observation vector



$$\boldsymbol{X_i}^{DF} = \begin{bmatrix} \overline{Z}^{m}_{K_u} \\ \overline{DFR}^{m}_{K_u - K_a} \\ \overline{RT} \end{bmatrix} \tag{4}$$

includes the averaged DF variables $\overline{Z}^{m}_{K_u}$ and $\overline{DFR}^{m}_{K_u - K_a}$, together with $\overline{RT}$. The ensuring description of the algorithm refers to an observation vector $\boldsymbol{X_i}$ and is valid for both multidimensional vectors $\boldsymbol{X}_i^{DF}$ and $\boldsymbol{X}_i^{DP}$.

In order to derive $p_k$, weighted centroids

$$\boldsymbol{\mu}_k = \frac{\sum_{i=1}^{n} w_i \boldsymbol{X_i}}{\sum_{i=1}^{n} w_i} \tag{5}$$

and weighted covariance matrices

$$C_k = \frac{\sum_{i=1}^{n} w_i \left( (\boldsymbol{X_i} - \boldsymbol{\mu_k})(\boldsymbol{X_i} - \boldsymbol{\mu_k})^{\top} \right)}{\sum_{i=1}^{n} w_i} \tag{6}$$

are calculated with the weighting factors $w_i = \text{qHPR}_k$ based on all available $S_i$ for each hydrometeor class k in DP and DF space. Besic et al. (2016) and Trömel et al. (2023) apply a clustering algorithms to the multidimensional DP measurements and identified clusters are then assigned to specific hydrometeor classes using state-of-the-art HMC. Centroids $\boldsymbol{\mu}_k$ (in Trömel et al. (2023) also $\boldsymbol{C}_k$) are then calculated for these clusters. However, non-physical clusters in terms of precipitation microphysics

and strict boundaries between clustered data may arise, which have an impact on the calculations of centroids and covariance matrices in polarimetric space. Instead, the use of qHPR as weighting factors enables a more physical transition between the DP or DF variables for different hydrometeor classes. The multidimensional distribution functions $p_k$ are calculated based on centroids $\boldsymbol{\mu}_k$ and covariances $\boldsymbol{C}_k$ (Eq.5 and Eq.6) assuming a multivariate normal distribution

$$p_k(\boldsymbol{X}|\boldsymbol{\mu_k}, \boldsymbol{C_k}) = \Lambda \exp\left( -\frac{1}{2}(\boldsymbol{X} - \boldsymbol{\mu_k})^{\top} \boldsymbol{C_k}^{-1}(\boldsymbol{X} - \boldsymbol{\mu_k}) \right) \tag{7}$$

with the transpose of a matrix $(\cdot)^{\top}$, the dimension $d$ of the multivariate normal distribution and $\Lambda = 1/\sqrt{(2\pi)^d |\boldsymbol{C_k}|}$, where $|\cdot|$ denotes the the determinant (Trömel et al., 2023). The multivariate normal distribution $p_k$ replaces the exponential distribution used in Besic et al. (2018), allowing a more suitable elliptical (instead of only spherical) distributions of DP or DF variables for different hydrometeor classes. Besic et al. (2018) use the entropy to determine the shape of $p_k$, which is a purely statistical method. The inherent assumption is that the entropy and thus the mixing is highest exactly between two centroids. Trömel

et al. (2023) describe the shape of $p_k$ with the observed distribution of the DP measurements in multidimensional space using the covariance matrices. Including now the qHPRs, as weighted factors, the centroids and covariance matrices are no longer restricted to the clusters with strict boundaries in polarimetric space, instead overlapping distributions are enabled.

The value of a $p_k(\boldsymbol{\mu_k})$ equals 1 according for an unmixed observation (Besic et al., 2018), of only one specific hydrometeor class. Therefore, each $p_k(\boldsymbol{X})$ is normalized with $p_k(\boldsymbol{\mu_k})$:

$$\tilde{p}_k = \frac{p_k(\boldsymbol{X})}{p_k(\boldsymbol{\mu_k})}. \tag{8}$$





Finally, HPRs for different hydrometeor classes k are estimated as follows:

$$\text{HPR}_k = \frac{W_k(T)\,\tilde{p}_k}{\sum_{k=1}^{n} W_k(T)\,\tilde{p}_k}. \tag{9}$$

The weighting functions $W_k(T)$ suppress HPR estimates of hydrometeor classes in unexpected temperature regions. $W_k(T)$ are derived from statistics of the relative occurrence of the different hydrometeor classes $(N(\text{HM}_k))$ in 2°C intervals between

-80°C and 32°C. Resulting estimates of partitioning ratios for different hydrometeor classes k are referred to as $\text{HPR}_k^{\text{DP}}$ and $\text{HPR}_k^{\text{DF}}$ in DP and DF space, respectively.

## 4    Results

### 4.1    Multidimensional distribution function $p_k$ in polarimetric and dual-frequency space

The $\tilde{p}_k$ of the DP (Fig. 4) and DF variables (Fig. 5d, e and f), as well as of RT (Fig. 5a, b and c) for each hydrometeor class

k are derived based on the training data set (Fig. 3, center) as described in Sect. 3 and represent the basis for the $\text{HMC}_\text{P}$. The hydrometeor classes are analyzed separately in the regions where they are most likely to occur, e.g. light rain (LR), moderate rain (MR), heavy rain (HR) and big drops (BD) in the liquid region, plates/dendrites (PD), ice crystals (IC) and snow (SN) in the solid region and wet snow (WS), graupel (GP), hail (HA) and rain/hail (RH) in the solid, liquid and melting region (mixed). Note that the figures mentioned above illustrate only two-dimensional representations of the multidimensional $\tilde{p}_k$, normalized

according to Eq. 8 without weighting $W_k(T)$.

In DP space, the centroids of the hydrometeor classes LR, MR, HR (Fig. 4a, d and g) show in the $Z_H$-$Z_{DR}$ and $Z_H$-$K_{DP}$ plane an increasing $Z_H$ with increasing $Z_{DR}$ and $K_{DP}$ respectively . With increasing $Z_H$ $\rho_{HV}$ is decreasing due to droplet growth and the associated increase in droplet flattening in liquid precipitation (Straka et al., 2000). The BD centroid shows an increased $Z_{DR}$ compared to LR, MR and HR (Bechini and Chandrasekar, 2015). In DP space of the solid region (Fig. 4b, e and h), the

centroids of PD differ from IC with respect to high $Z_{DR}$ and $K_{DP}$ values, which is in line with expected characteristics of ice particles especially in the dendritic growth layer (DGL). SN instead is characterized by reduced $Z_{DR}$ and $K_{DP}$ values but higher $Z_H$ values, which is due to the increase in particle size and decrease in density during aggregation processes. As expected the $\tilde{p}_k$ for PD show reduced $\rho_{HV}$ values due to the diversity of ice particles in the DGL (Trömel et al., 2019; Thompson et al., 2014). With regard to the mixed hydrometeors (Fig. 4c, f and i) $Z_H$ of the WS centroid is much lower compared to the ones of

GP, RH and HA. The latter shows the highest $Z_H$. $\rho_{HV}$ is the lowest in WS, followed by the two hail classes and then GP. HA and GP show lower $Z_{DR}$ values compared to RH and WS due the impact of tumbling of HA and potentially conical shapes of GP (Straka et al., 2000).

In DF space (only $Z_{K_u}^m$-$\text{DFR}_{K_u-K_a}^m$ space is shown) the centroids for liquid hydrometeors (Fig. 5d) show the typical behavior with increasing $\text{DFR}_{K_u-K_a}^m$ and increasing $Z_{K_u}^m$ due to increasing attenuation effects transitioning from LR to HR (Le and

Chandrasekar, 2012). An even more pronounced increase in $\text{DFR}_{K_u-K_a}^m$ with rising $Z_{K_u}^m$ is observed for BD. This can be attributed to the additional influence of non-Rayleigh scattering effects when the droplet diameter exceeds 0.8 mm (Mroz et al.,






**Figure 4.** Normalized probability density functions $\tilde{p}_k$ of the DP variables $Z_H$ against $Z_{DR}$ (a, b, and c), $Z_H$ against $K_{DP}$ (d, e and f) and $Z_H$ against $\rho_{HV}$ (g, h and i) for liquid hydrometeors (LR, MR, HR and BD, left column), solid hydrometeors (PD, SN, IC, center column) and mixed phase hydrometeors (RH, WS, GP and HA, right column). The different contour lines indicating the probabilities of the given $\tilde{p}_k$ for the different hydrometeor classes.



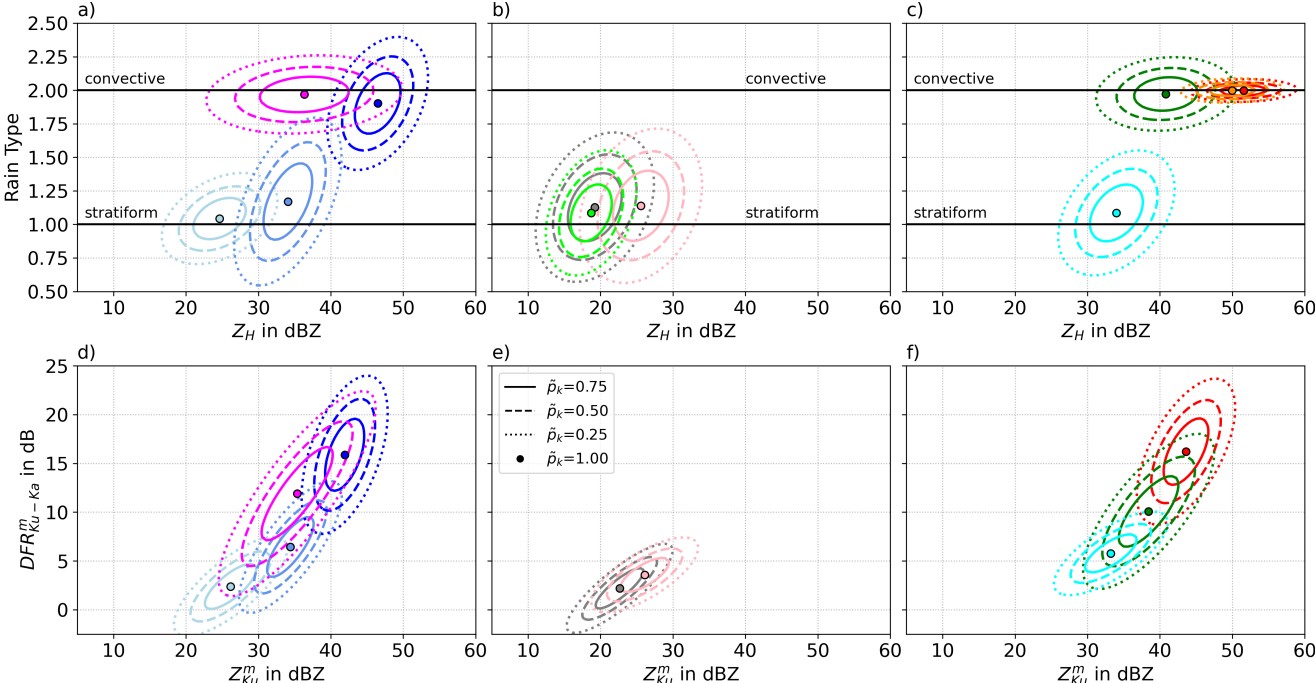

**Figure 5.** Normalized probability density function $\tilde{p}_k$ of $Z_H$ in relation to the RT (a, b and c) and the DF variables ($Z_{K_u}^m$ in relation to $DFR_{K_u-K_a}^m$, d, e and f), for liquid hydrometeors (LR, MR, HR and BD, left column), solid hydrometeors (PD, SN, IC, center column) and mixed phase hydrometeors (RH, WS, GP and HA, right column). The different contour lines indicating the probabilities of the given $\tilde{p}_k$ for the different hydrometeor classes.

2024). $DFR_{K_u-K_a}^m$ show an increase for both IC and SN with increasing $Z_{K_u}^m$ due to the increasing impact of non-Rayleigh effects with increasing particle diameter (Fig. 5e). GP, WS and RH (Fig. 5f) show increased $DFR_{K_u-K_a}^m$ due to a combination of increased diameters, riming and attenuation effects (Le and Chandrasekar, 2021a; Tridon et al., 2019). Note that DF variables
do not significantly differ between RH and HA nor between IC and PD (not shown). As a consequence the hydrometeor classes are merged to RH and IC.

BD, GP, RH and HA are mostly restricted to convective precipitation where HR has higher tendency to appear also in stratiform precipitation. LR, WS, SN, IC and PD are restricted to stratiform precipitation where WS, SN, IC and PD can occur with lower probabilities also in convection (Fig. 5a, b and c).

**4.2 Evaluation with quasi hydrometeor partitioning ratios**

In order to evaluate the DF- and DP-based HPR retrievals, $HMC_P$ estimates (E) of the test dataset are compared to the qHPR serving as the reference (R), with the following statistical metrics:





$$\text{BIAS} = \sqrt{\frac{1}{N}\sum_{i=1}^{N}\left(E_i - R_i\right)} \quad, \tag{10}$$

$$\text{RMSE} = \sqrt{\frac{1}{N}\sum_{i=1}^{N}\left(E_i - R_i\right)^2} \text{ and} \tag{11}$$

$$\text{CCP} = \frac{\sum_{i=1}^{N}\left(E_i - \overline{E}\right)\left(R_i - \overline{R}\right)}{\sqrt{\sum_{i=1}^{N}\left(E_i - \overline{E}\right)^2 \sum_{i=1}^{N}\left(R_i - \overline{R}\right)^2}}. \tag{12}$$

$\overline{\text{R}}$ and $\overline{\text{E}}$ denote the mean values of $\text{R}_i$ and $\text{E}_i$, respectively. A comparison between the qHPRs and HPRs based on the $\overline{\text{DF}}$ and $\overline{\text{DP}}$ variables results in high CCP for several hydrometeor classes. E.g., CCPs higher than 0.8, are achieved with the DP-based retrievals for LR, MR, HR, WS, IC and SN and with the DF-based retrievals for LR, MR, and SN (Fig. 6a, b, c, d, e, o, m, q and r). The lowest correlations occur with $\text{HPR}_{\text{BD}}^{\text{DP}}$ and $\text{HPR}_{\text{BD}}^{\text{DF}}$ (Fig. 6g and h) followed by $\text{HPR}_{\text{RH}}^{\text{DF}}$, $\text{HPR}_{\text{PD}}^{\text{DP}}$ and $\text{HPR}_{\text{IC}}^{\text{DF}}$ (Fig.

6j, n and t). The largest underestimations can be found in snow in both the DP and DF space, with a BIAS up to -5.15% (Fig.6, q and r) followed $\text{HPR}_{\text{LR}}^{\text{DP}}$, $\text{HPR}_{\text{MR}}^{\text{DP}}$, $\text{HPR}_{\text{IC}}^{\text{DF}}$ and $\text{HPR}_{\text{WS}}^{\text{DF}}$(Fig.6, a, c, n and p). Pronounced HPR overestimation occurs for HR, GP for DF- and DP-based retrievals as well as for $\text{HPR}_{\text{BD}}^{\text{DP}}$ (Fig. 6e, f, g, k, and l). The comparison of hail HPRs shows a overestimation of $\text{HPR}_{\text{RH}}^{\text{DP}}$ and $\text{HPR}_{\text{HA}}^{\text{DP}}$ and small underestimation of $\text{HPR}_{\text{RH}}^{\text{DF}}$ (Fig. 6i, j and s). Note that qHPR estimated from the dominant hydrometeor classes may overestimate the actual partitioning ratios due to the disproportional impact of

hail on DP variables. As a consequence the biases in $\text{HPR}_{\text{RH}}^{\text{DP}}$ and $\text{HPR}_{\text{HA}}^{\text{DP}}$ may be even more pronounced than indicated by the qHPR-based evaluation. BIAS and RMSE values are small for the hail classes and BD, which can be attributed to their overall low HPR values.

In summary the DP-based retrievals outperform the ones based on DF in terms of CCP and RMSE, in most cases also with respect to the BIAS values. This can be attributed to the higher information content of DP compared to DF measurements, for

example, regarding the shape, orientation and homogeneity of the hydrometeors within the measurement volume. Except for the BD estimates, the retrievals for liquid hydrometeors in both DF- and DP-space, achieve a higher accuracy compared to the retrievals for the solid hydrometeor classes, reflecting the increased complexity and variability of DP and DF signals for solid and mixed hydrometeors.

## 4.3 Case study

To verify and illustrate the plausibility of the $\text{HMC}_{\text{P}}$ retrievals, a GPM overflight is directly compared to the KDDC NEXRAD GR and a pseudo range height indicator (RHI) is generated along DPRs along-track scan (Fig. 7a, blue dashed and red solid lines). The DP (Fig. 7b, d, f and h) and DF (Fig. 7c, e and g) variables are exploited to derive and compare the $\text{HPR}_{\text{k}}^{\text{DP}}$(Fig. 9) and $\text{HPR}_{\text{k}}^{\text{DF}}$(Fig. 8) with the $\text{HMC}_{\text{P}}$.







**Figure 6.** Two-dimensional histograms of the pairwise comparison $HPR_k^{DF}$ and $HPR_k^{DP}$ with $qHPR_k$ for the different hydrometeor classes. The CCP, BIAS and RMSE are in black, blue and red. The colors indicate the count of samples and the black solid line the 1:1 relationship.





A comparison of the GR and SR measurements (Fig. 7) reveals a slight discrepancy in the STH. While GR measurements indicate a STH of approximately 15 km, SR indicate lower values due to the $K_u$PR and $K_a$PR sensitivity (Iguchi et al., 2010). The precipitation event can be subdivided into a stratiform and convective region. According to the GR-based RT classification, the convective area starts at a distance of 60 km from the GR, while the SR classification indicates that the convective area starts at a distance of 50 km. This discrepancy is likely attributable to the presence of a bright band located approximately at 3.5 km height characterized by increased $Z_H$, $Z_{DR}$, and reduced $\rho_{HV}$ values, which is not properly identified by the DPR between 50 km and 60 km range and thus partly classified as a convective region. Additionally, the GR has identified further convective areas up to a distance of 40 km, which may not be detected by SR due to their relatively small scale. The GR beams of higher elevation angles are affected by differential attenuation in the ML resulting in negative $Z_{DR}$ (Fig. 7d). At distances of approximately 80 km and beyond, the measurements at low elevation angles are partially affected by non-uniform beam filling (NBF), characterized by extreme high $\phi_{DP}$ (not shown) and low $\rho_{HV}$ values (Ryzhkov and Zrnic, 2019). In the convective region, the impact of attenuation on SR near surface measurements is significant, especially at $K_a$-band (not shown). The signal partially drops below the 18 dBZ $K_a$-band threshold and is therefore excluded.

Until 50 km range, enhanced $Z_{DR}$ and $DFR^m_{K_u-K_a}$ and moderate $Z^m_{K_u}$ and $Z_H$ values result in corresponding increased $HPR^{DF}_{MR}$ and $HPR^{DP}_{MR}$ and low $HPR^{DF}_{LR}$ and $HPR^{DP}_{LR}$ (Fig. 8a, b and Fig. 9a, b). $HPR^{DF}_{BD}$ appears with low ratios in the convective region, whereas $HPR^{DP}_{BD}$ does not show a clear signal. In a range between 10 km to 30 km the small-scale convective regions are not detected by the DPR resulting in no $HPR^{DF}_{BD}$ where small proportions of $HPR^{DP}_{BD}$ are still estimated (Fig. 8d and Fig. 9d). DP estimates effectively illustrate the transition from solid hydrometeors such as SN via WS to liquid hydrometeors such as LR, MR and HR. However, $HPR^{DF}_{WS}$ do not match with DPRs bright-band detection where $HPR^{DP}_{WS}$ is restricted between $T_b = 0°C$ and $T_t = 4°C$ (Fig. 8h and Fig. 9h). HR is apparent in DP measurements within the ML, which is not the case in the DF measurements (Fig. 8c and Fig. 9c).

Both DF and DP measurements allocate the transition zone from ice to snow retrievals at approximately 8 km altitude, which corresponds to the height of the DGL (Fig. 8g, i and Fig. 9g, i) identified by increased $K_{DP}$ values slightly above the $-15°C$ isotherm (Fig. 7f). In the measurements obtained at ranges up to 20 km a decrease in $K_{DP}$ and an increase in $Z_H$ can be identified below the $-15°C$ isotherm (Fig. 7b, d and f) indicating aggregation processes (Trömel et al., 2019). This is also supported by increasing $DFR^m_{K_u-K_a}$ measurements in the same region (Fig. 7g). Increased snow HPRs above the $-15°C$ isotherm may be connected to the underestimation of ice HPRs, as identified in Sect.4.2. The partial occurrence of $HPR^{DP}_{PD}$ in the DGL (Fig. 9j) is challenging to interpret due to the differential attenuation (Fig. 7d).

As expected rimed hydrometeors like GP, RH and HA, are primarily observed in convective regime. Due to the discrepancy between RT classifications based on GR and SR measurements (f in Fig. 8 and Fig. 9) high $HPR^{DF}_{GP}$ extend over a larger region compared to $HPR^{DP}_{GP}$. Overall HPR of GP in DP and DF are significantly overestimated (compare Sect.4.2). A comparison of hail HPRs reveal a comparable vertical distribution up to an altitude of approximately 8 km. Note that $HPR^{DP}_{RH}$ and $HPR^{DP}_{HA}$ (Fig. 9e and k) have to be considered combined for a direct comparison with $HPR^{DF}_{RH}$ (Fig. 7e). In regions with NBF, the detection of hail has to be considered with caution due the similarity of the DP signals for NBF and hail. However, SR



partially confirms hail HPR in these areas. Due to the overall overestimation (underestimation) of hail HPRs in DP (DF) space, according estimates should be treated with caution.

## 5 Conclusions

This paper describes the most recent improvements of a more sophisticated hydrometeor classification (HMC) scheme to derive also hydrometeor partitioning ratios (HPRs). Such an algorithm has been first introduced by Besic et al. (2018) and enhanced in Trömel et al. (2023) ($\mathrm{HMC_P}$). $\mathrm{HMC_P}$ is capable to derive HPRs from dual-polarization (DP) measurements ($\mathrm{HMC_P^{DP}}$) of ground-based radars (GR) for each resolved volume. Combining GR DP observations from NEXRAD's WSR-88D S-band radars with space-borne radar (SR) dual-frequency (DF) observations, more precisely from the Dual-Frequency Precipitation Radar (DPR) onboard the Global Precipitation Measurement core satellite (GPM), allows to extend $\mathrm{HMC_P}$ for DF-based HPR estimates from SR observations ($\mathrm{HMC_P^{DF}}$). Matching SR and GR observations, superobbed volumes containing a large number of GR pixels are generated and enable the estimation of quasi HPRs (qHPRs). These qHPRs represent the hydrometeor mixtures in superobbed volumes and are calculated with the identified dominant hydrometeor classes applying the modified standard HMC to the high-resolution GR measurements. The averaged DF and DP variables and qHPRs of the supperobbed volumes are exploited for the training of $\mathrm{HMC_P}$ and also for the ensuing evaluation of HPR estimates. Such estimates are either based on DP ($\mathrm{HPR_k^{DP}}$) or DF ($\mathrm{HPR_k^{DF}}$) observations and compared with the qHPRs derived from averaged DF and DP variables, respectively. The derived $\tilde{p}_k$, which form the basis for the $\mathrm{HMC_P}$, are in line with expected DP and DF observations for different hydrometeor classes (e.g. Straka et al., 2000; Bechini and Chandrasekar, 2015; Trömel et al., 2019; Thompson et al., 2014). A comparison between qHPRs and HPRs in DF ($\mathrm{HPR_k^{DF}}$) and DP-space ($\mathrm{HPR_k^{DP}}$) results in correlations higher than 0.7 for various hydrometeor classes. Lowest correlations are obtained for big drops in both DP- and DF-space with 0.38 and 0.15, respectively, followed by correlations for $\mathrm{HPR_k^{DF}}$ of ice with 0.59, dendrites/plates and rain/hail both with 0.56. $\mathrm{HMC_P}$ overestimates graupel and underestimates snow HPRs in DF and DP space. Hail HPRs are overestimated in DP and slightly underestimated in DF space. Overall, HPR estimates are more accurate in DP space than in DF space and perform best for liquid hydrometeors, except for big drops. DP observation provide additional information e.g. on the shape, orientation and homogeneity of the hydrometeors within the measurement volume compared to DF observations leads to more accurate derivations of HPR. Furthermore, $\mathrm{HMC_P^{DP}}$ and $\mathrm{HMC_P^{DF}}$ have been trained with DP data, also promoting a better performance in DP space. A case study revealed a high degree of agreement between GR- and SR-based estimates as well as a plausible vertical distribution of HPRs in the light of the DF and DP measurements.

Including additional information in the multidimensional observation vectors $\boldsymbol{X}_i^{\mathrm{DF}}$ and $\boldsymbol{X}_i^{\mathrm{DP}}$ could further improve the accuracy of the HPR estimates. E.g., for $\boldsymbol{X}_i^{\mathrm{DF}}$ vertical gradients of $\mathrm{Z_{K_u}^m}$ or $\mathrm{DFR_{K_u-K_a}^m}$ can be exploited. Battaglia et al. (2014), Mroz et al. (2018) and Le et al. (2016) demonstrated already their information content for the detection of hail and wet snow. Observations from other satellite devices, e.g. brightness temperatures from GPMs passive microwave radiometer utilized for hail (Mroz et al., 2017) or snow (Rysman et al., 2018, 2019) detection, could also be exploited to increase the information





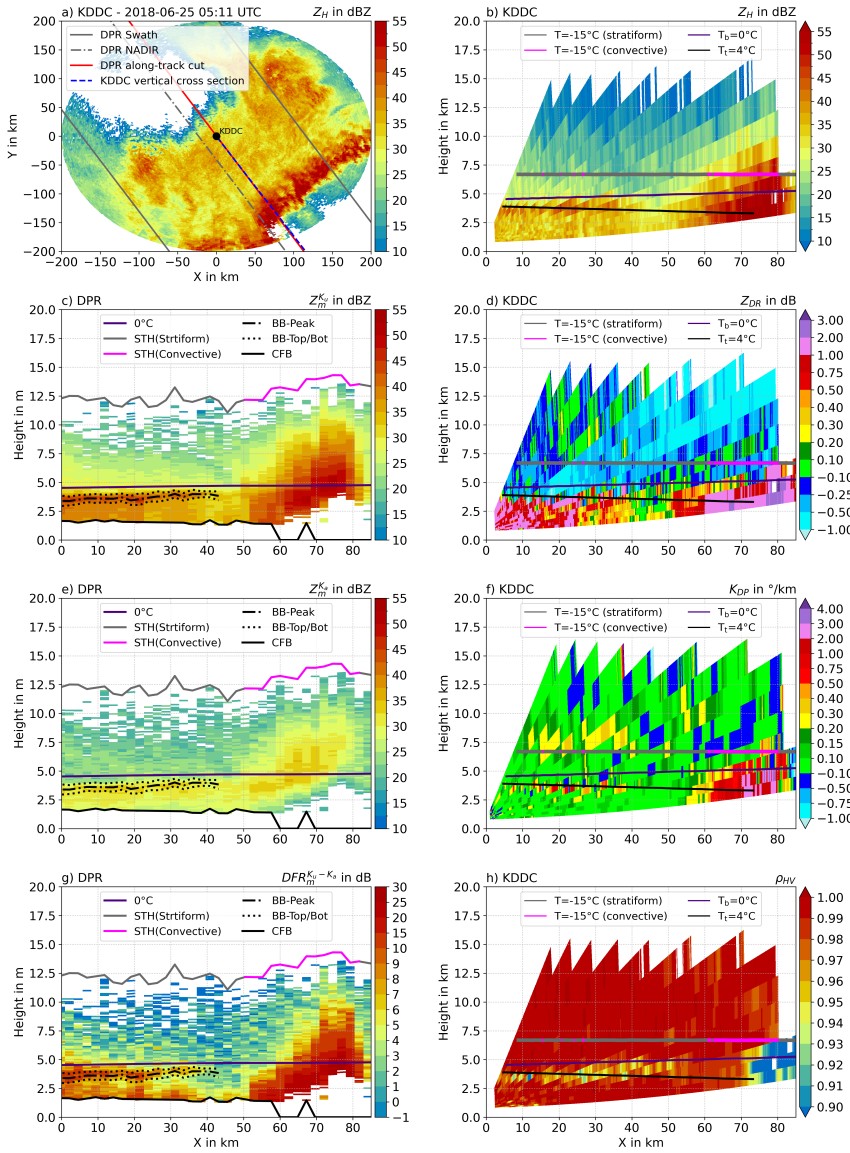

**Figure 7.** a) PPI of $Z_H$ measured on 25 June 2018 at 05:11 UTC with KDDC and overpassed by GPM (orbit number 024557). Nadir along-track vertical cut of DPR-observed $Z_{Ku}^m$ (c), $Z_{Ka}^m$ (e) and $DFR_{Ku-Ka}^m$ (g). Pseudo RHIs of $Z_H$ (b), $Z_{DR}$ (d), $K_{DP}$ (f), $\rho_{HV}$ (h) along DPR's vertical cut. In panel a the gray lines indicate DPRs outer swath, the gray dashed line DPRs NADIR scan, the red line the along-track vertical cut and the blue line the location of the vertical cross section of the GR. In panels c, e and g the black line indicates the clutter free bottom, the indigo line the freezing level height (DPR), and the STH for convective (magenta) and for stratiform (cyan) SR-based RT. The dashed lines represent the bright band top and bottom. The bright band peak is illustrated as dash-dotted line. In panels b, d, f and h the black lines indicate the $T_t = 4°C$ and the indigo lines the $T_b = 0°C$ isotherms. The $T = -15°C$ is indicated in gray/magenta for the GR-based stratiform/convective RT.





**Figure 8.** Estimated $\mathrm{HPR}_k^{\mathrm{DF}}$ for different hydrometeor classes applying $\mathrm{HMC}_P$ to SR observations shown in Fig. 7. The black line indicates the clutter free bottom, the indigo line the freezing level height (DPR), and the STH for convective (magenta) and for stratiform (gray) SR-based RT. The dashed lines represent the bright band top and bottom. The bright band peak is illustrated as dash-dotted line.



**Figure 9.** Estimated $\mathrm{HPR}_k^{\mathrm{DP}}$ for different hydrometeor classes with $\mathrm{HMC_P}$ with GR observations shown in Fig. 7 The black lines indicate the $\mathrm{T_t} = 4°C$ and the indigo lines the $\mathrm{T_b} = 0°C$ isotherms. The $\mathrm{T_c} = -15°C$ is indicated in gray/magenta for the GR-based stratiform/convective RT.





content. With respect to GR observations, the depolarization ratio, which has been shown to be valuable for riming detection (Blanke et al., 2024), might extend $\boldsymbol{X}_\mathrm{i}^\mathrm{DP}$.

    The retrievals introduced in this paper can be considered as valuable for different meteorological aspects. E.g., more accurate hydrometeor classifications can refine the calibration of GR with SR observations (Cao et al., 2013; Pejcic et al., 2022) by adapting the frequency transformation much more precisely to specific hydrometeor classes. Using $\mathrm{HMC}_\mathrm{P}^\mathrm{DF}$, the GPM DPRs

area-wide measurements now provide precise information on hydrometeor distributions in areas without GR measurements. This allows e.g. to extend the evaluation of hydrometeor distributions in numerical weather prediction (NWP) models (Trömel et al., 2023) to the global scale. The assimilation of GR-based measurements and retrievals (Trömel et al., 2023; Reimann et al., 2023), but also of SR-based reflectivity measurements (Ikuta et al., 2021; Kotsuki et al., 2023) and rainfall estimates Li et al. (2020) in NWP has been shown to improve the accuracy of numerical precipitation prediction. Thus, the assimilation of DF-

and DP-based HPRs may further improve the representation of hydrometeors in NWP.

*Code and data availability.* Codes for the data processing and intermediate data products can be made available upon request. $\mathrm{HMC}_\mathrm{P}$ is available at $\omega$radlib.The GPM data can be downloaded following Iguchi and Meneghini (2021) and the quality-controlled ground radar data can be requested by NASA's GPM Ground Validation program (GPM-GV).

## Appendix A: The standard Hydrometeor Classification to identify the dominant hydrometeor class

The membership functions (MSFs, Tab. A1) for the hydrometeor classes heavy rain (HR), big drops (BD), rain/hail (RH), wet snow (WS) and graupel (GP) are adapted from Park et al. (2009), whereas the rain class is subdivided into the light rain (LR) and moderate rain (MR) classes with a $Z_\mathrm{H}$ threshold of 28 dBZ (Tab. A1 LR and MR columns) following Straka et al. (2000). Furthermore, the $Z_\mathrm{DR}$ MSF for IC is extended to negative values and the $Z_\mathrm{H}$ MSF for SN and IC includes the snow ice switch-over between 15 dBZ and 20 dBZ (Tab. A1 SN and IC column) following Thompson et al. (2014).

PD is added as a new class combining the MSFs of plates and dendrites from Thompson et al. (2014). For this purpose, the MSFs of the two hydrometeor classes are superimposed and only the outer boundaries are considered (Tab. A1 PD column). The $\rho_\mathrm{HV}$-MSF for IC are used also for PD. For HA, the polarimetric MSF from Dolan et al. (2013) are applied. In general the trapezoidal MSFs of HA and PD are tuned until they overlap as good as possible with the membership beta functions used in Dolan et al. (2013) and Thompson et al. (2014).

The temperature MSFs are designed to allow solid phase hydrometeors IC and SN only at temperatures below 0°C and liquid phase hydrometeors at temperatures above 0°C. WS and PD are restricted to temperature regimes with their highest probability of occurrence (von Terzi et al., 2022; Lundquist et al., 2008; Heymsfield et al., 2021) and hydrometeors such as RH, BD, HA and GP are allowed to exist in all regions (liquid, solid and mixed phase). BD are restricted up to -32.5°C assuming the 6.5°C/km lapse rate. This corresponds to findings of van Lier-Walqui et al. (2016) reporting updrafts reaching around 5 km



above the freezing level. GP MSF for temperature are set to the temperature interval between -50 $^\circ C$ and 30 $^\circ C$, which is consistent with the boundaries for high density and low density graupel in Dolan et al. (2013).

**Table A1.** Values x1, x2, x3 and x4 of the used trapezoidal membership functions for $Z_H$, $Z_{DR}$, $LK_{DP}$, $\rho_{HV}$ and T. f1, f2, f3, g1 and g2 can be found in Park et al. (2009) Eq.4 and Eq.5

|  | LR | MR | HR | BD | RH | GP | CR | DS | WS | PD | DH |
|---|---|---|---|---|---|---|---|---|---|---|---|
| x1($Z_H$) in $dBZ$ | **5.0** | **23.0** | 40.0 | 20.0 | 45.0 | 25.0 | 0.0 | **15.0** | 25.0 | **-1.0** | **45.0** |
| x2($Z_H$) in $dBZ$ | **10.0** | **28.0** | 45.0 | 25.0 | 50.0 | 35.0 | 5.0 | **20.0** | 30.0 | **2.0** | **50.0** |
| x3($Z_H$) in $dBZ$ | **28.0** | **45.0** | 55.0 | 45.0 | 75.0 | 50.0 | **15.0** | 35.0 | 40.0 | **26.0** | **67.0** |
| x4($Z_H$) in $dBZ$ | **33.0** | **50.0** | 60.0 | 50.0 | 80.0 | 55.0 | **20.0** | 40.0 | 50.0 | **31.0** | **72.5** |
| x1($Z_{DR}$) in $dB$ | f1-0.3 | f1-0.3 | f1-0.3 | f2-0.3 | -0.3 | -0.3 | **-1.0** | -0.3 | 0.5 | **1.3** | **-0.5** |
| x2($Z_{DR}$) in $dB$ | f1 | f1 | f1 | f2 | 0.0 | 0 | **-0.8** | 0.0 | 1.0 | **1.6** | **-0.25** |
| x3($Z_{DR}$) in $dB$ | f2 | f2 | f2 | f3 | f1 | f1 | 3.0 | 0.3 | 2.0 | **8.4** | **0.50** |
| x4($Z_{DR}$) in $dB$ | f2+0.5 | f2+0.5 | f2+0.5 | f3+1 | f1+0.5 | f1+0.5 | 3.3 | 0.6 | 3.0 | **9.2** | **0.75** |
| x1($LK_{DP}$) | g1-1 | g1-1 | g1-1 | g1-1 | -10.0 | -30.0 | -5.0 | -30.0 | -30.0 | **-30.0** | **-30.0** |
| x2($LK_{DP}$) | g1 | g1 | g1 | g1 | -4.0 | -25.0 | 0.0 | -25.0 | -25.0 | **-13.0** | **-29.0** |
| x3($LK_{DP}$) | g2 | g2 | g2 | g2 | g1 | 10.0 | 10.0 | 10.0 | 10.0 | **-3.0** | **4.8** |
| x4($LK_{DP}$) | g2+1 | g2+1 | g2+1 | g2+1 | g1+1 | 20.0 | 15.0 | 20.0 | 20.0 | **2.15** | **7.0** |
| x1($\rho_{HV}$) | 0.95 | 0.95 | 0.92 | 0.92 | 0.85 | 0.90 | 0.95 | 0.95 | 0.88 | **0.94** | **0.80** |
| x2($\rho_{HV}$) | 0.97 | 0.97 | 0.95 | 0.95 | 0.90 | 0.97 | 0.98 | 0.98 | 0.92 | **0.97** | **0.91** |
| x3($\rho_{HV}$) | 1.00 | 1.00 | 1.00 | 1.00 | 1.00 | 1.00 | 1.00 | 1.00 | 0.95 | **0.99** | **0.99** |
| x4($\rho_{HV}$) | 1.01 | 1.01 | 1.01 | 1.01 | 1.01 | 1.01 | 1.01 | 1.01 | 0.985 | **1.00** | **1.00** |
| x1($T$) in $^\circ C$ | **0.0** | **0.0** | **0.0** | -32.5 | -90.0 | -50.0 | -90.0 | -90.0 | -2.0 | -20.0 | -90.0 |
| x2($T$) in $^\circ C$ | **1.0** | **1.0** | **1.0** | -19.5 | -50.0 | -40.0 | -80.0 | -80.0 | 0.0 | -17.5 | -50.0 |
| x3($T$) in $^\circ C$ | **50.0** | **50.0** | **50.0** | 50.0 | 50.0 | 5.0 | -2.0 | -2.0 | 4.0 | -12.5 | 0.0 |
| x4($T$) in $^\circ C$ | **55.0** | **55.0** | 55 | 55.0 | 55.0 | 30.0 | 0.0 | 0.0 | 6.0 | -10.0 | 5.0 |

*Author contributions.* VP developed the methodology for this work, designed the manuscript, performed the coding, processed the data and carried out the visualization and analysis. KMu supported the code for the VMM. ST and KMr provided the scientific advice and support in the development of the text.

*Competing interests.* The authors state that they have no conflict of interest.



*Acknowledgements.* Velibor Pejcic's research was carried out partially in the framework of the priority programme SPP 2115 "Polarimetric Radar Observations meet Atmospheric Modelling (PROM)" within the project "Operation Hydrometeors" and the research project Near-Realtime Precipitation Estimation and Prediction (RealPEP). Work done by Kamil Mroz was performed under a contract with the National Centre for Earth Observation. We would also like to extend our gratitude to Jason Pippitt and Daniel Watters for providing and supporting us with the GPM GV GR data and also thank NASA/JAXA for providing the GPM DPR data. Furthermore, we would like to express our gratitude to Julian Steinheuer for his scientific support.

*Financial support.* This research was funded by the German Research Foundation (Deutsche Forschungsgemeinschaft, DFG; 320397309 and 408027387).



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
