# Peer review of "Hydrometeor partitioning ratios for dual-frequency space-borne and polarimetric ground-based radar observations"

_EGUsphere, 2025_

## Author Comment (AC1)

Revision of : Hydrometeor partitioning ratios for dual-frequency space-borne and polarimetric ground-based radar observations by Velibor Pejcic, Kamil Mroz, Kai Mühlbauer, and Silke Trömel

**Response to reviewer RC1**

Dear reviewer,

We are very grateful for all your suggestions to further improve the manuscript. Both reviewers suggested reducing the number of abbreviations to improve readability. We agree trying to improve the readability and reduce the number of abbreviations in the manuscript. E.g., we decided to remove all abbreviations for the hydrometeor classes (reducing the number by 11 abbreviations). We also changed the individual abbreviations of the hydrometeor partitioning ratios for each specific hydrometeor class in the manuscript. For example, the hydrometeor partitioning ratio for light rain derived from dual-frequency measurements is no longer abbreviated as HPR\_LR^DF but is now referred to as HPR\_DF of light rain. This reduced the number of abbreviations by an additional 20 (11 HPR based on dual-polarisation and 9 HPR based on dual-frequency). Abbreviations for multi-scattering (MS) and drop size distributions (DSD) are removed as well as they do not appear frequently in the text. Furthermore, we simplified the complex abbreviation DFR^m\_(Ku-Ka) to DFR. We also included a list of all remaining abbreviations for the reader's convenience ( see Table C1 in Appendix C) and added "All abbreviations can be found in Tab. C" in line 62.

Additionally, we like to draw your attention to the fact that former Figures 8 and 9 (now Figures 7 and 8 in the revised manuscript) changed slightly. Unfortunately, an earlier version of the centroids and covariances was accidentally used for the case study run (plausibility check), i.e. not the ones presented in our manuscript (see Fig. 4) and used for training and evaluation. The small differences do not affect our overall results and conclusions. We apologize for this minor error.

Figure 7. Estimated HPR^DF\_k for different hydrometeor classes applying HMC\_P to SR observations shown in Fig. 6. The black line indicates the clutter free bottom, the indigo line the freezing level height (DPR), and the STH for convective (magenta) and for stratiform (gray) SR-based RT. The dashed lines represent the bright band top and bottom. The bright band peak is illustrated as dash-dotted line

Figure 8. Estimated HPR^DP\_k for different hydrometeor classes with HMC\_P with GR observations shown in Fig. 6 The black lines indicate the Tt =  $4^{\circ}$ C and the indigo lines the Tb =  $0^{\circ}$ C isotherms. The Tc =  $-15^{\circ}$ C is indicated in gray/magenta for the GR-based stratiform/-convective RT.

Furthermore we updated the doi of the citation "A new aggregation and riming discrimination algorithm based on polarimetric weather radars" and we introduced the missing rain type RT in line 99.

Our response is highlighted in blue below. The revised manuscript with tracked changes is also provided for better transparency.

**Response to reviewer 1:**

This is a study of precipitation particle type classification and presents useful results. However, it is difficult to read due to too many abbreviations.

Line 3 It is unclear what the P in HMC^DP\_P means.

HMC\_P describes the more sophisticated hydrometeor classifications HMCs capable of deriving even hydrometeor partitioning ratios (HPRs). The subscript P points to the author's name "Pejcic" who advanced and evaluated these newer algorithms in this publication. We do not explain the association with the author's name in the abstract, but explain it later in line 50:

In the revised manuscript, the formulation in the abstract is changed to "Conventional radar-based hydrometeor classification algorithms identify the dominant hydrometeor type within a resolved radar volume, while newer techniques estimate the proportions of individual hydrometeor classes (hydrometeor partitioning ratios, HPRs) within a mixture. These newer algorithms (HMCDP) are based on dual-polarization measurements from ground-based radars (GR), while to date no comparable algorithms for space-borne radars (SR) with dual-frequency capabilities exist."

Later in Line 50 we clarify: "In this study, the HMC scheme from Trömel et al. (2023) (HMCP; introduced by Pejcic et al., 2021),..."

**Line 6-7 It is unclear what the k in HPR^DF\_k and HPR^DP\_k refers to.**

Subscript k stands for the k different hydrometeor classes considered in HMC\_P. However, we can introduce this subscript later and avoid it in the abstract. We rewrote in the abstract: "This study (1) further improves HPR estimates based on GR dual-polarization measurements, (2) exploits the combination of dual-frequency SR and dual-polarization GR to introduce HPRs based on dual-frequency observations only, and (3) evaluates GR- and SR-based HPR retrievals. " (Line 5)

Line 9 What does "these" refer to? there are no DP measurements in GPM/DPR.

It refers to the three objectives. We clarified and changed it in Line 7 to "To achieve these objectives...".

Line 37 The u and a in Ku-band and Ka-band are not subscripts.

Yes, thanks for pointing us to this mistake. It is now corrected throughout the manuscript.

Figure 2 Why are NEXRAD radar sites concentrated in the eastern half of the country?

The quality-controlled GR observations are provided by NASA's GPM Ground Validation program (GPM-GV), which only takes the shown radar sites into account. For more information we refer to Pippitt et al. (2013) listed in our manuscript. We added in Line 108: "Only quality-controlled observations provided by NASA's GPM Ground Validation program (GPM-GV) are used. The eastern GR sites of the NEXRAD network are predominantly used in the GPM-GV.". Furthermore, we edited line 109- "Quality-controlled GR observations are provided by NASA's GPM Ground Validation program (GPM-GV). The quality control includes non-precipitating echos removal with different thresholds and phase unfolding" is reformulated as follows: "GPM-GVs quality control includes the removal of non-precipitating echoes with different thresholds and phase unfolding." We also changed the caption of Figure 1: "Overview of the NEXRAD weather radar (WSR-88D) sites used in this study." to "NEXRAD weather radar (WSR-88D) sites provided and quality controlled by the GPM-GV and exploited in this study."

Line 64 The u and a in KuPR and KaPR are not subscripts.

Changed.

Line 66 November 2024 -> November 2023

Changed.

Line 95 It is unclear what MS indicates.

MS indicated "multiple scattering" but to reduce the amount of abbreviations in the manuscript it is no longer used in the revised version.

Line 149 Please explain what kind of particles are big drops (BD)

The hydrometeor class big drops originates from Park et al. (2009) and is now explained in the Appendix B where all hydrometeor classes are detailed. We added: "The big drops hydrometeor class originates from Park et al. (2009) and represents rain with a skewed drop size distribution towards larger raindrops, indicating the presence of raindrops with a diameter greater than 3 mm and a lack of smaller raindrops."

Line 171 training dataset -> test dataset

**Changed.**

Figure 4 A legend should be attached to each subfigure.

The legends in panels b) and c) are valid for all panels, but we agree that this is irritating. Instead, we suggest a superordinate legend for all plots instead of repeating the same legend in all panels.

Line 263 "an" overestimation

Changed.

---

## Author Comment (AC2)

Revision of : Hydrometeor partitioning ratios for dual-frequency space-borne and polarimetric ground-based radar observations by Velibor Pejcic, Kamil Mroz, Kai Mühlbauer, and Silke Trömel

**Response to reviewer 2 / editor EC1:**

Dear reviewer,

We are very grateful for all your suggestions to further improve the manuscript. Both reviewers suggested reducing the number of abbreviations to improve readability. We agree trying to improve the readability and reduce the number of abbreviations in the manuscript. E.g., we decided to remove all abbreviations for the hydrometeor classes (reducing the number by 11 abbreviations). We also changed the individual abbreviations of the hydrometeor partitioning ratios for each specific hydrometeor class in the manuscript. For example, the hydrometeor partitioning ratio for light rain derived from dual-frequency measurements is no longer abbreviated as HPR\_LR^DF but is now referred to as HPR\_DF of light rain. This reduced the number of abbreviations by an additional 20 (11 HPR based on dual-polarisation and 9 HPR based on dual-frequency). Abbreviations for multi-scattering (MS) and drop size distributions (DSD) are removed as well as they do not appear frequently in the text. Furthermore, we simplified the complex abbreviation DFR^m\_(Ku-Ka) to DFR. We also included a list of all remaining abbreviations for the reader's convenience ( see Table C1 in Appendix C) and added "All abbreviations can be found in Tab. C" in line 62.

Additionally, we like to draw your attention to the fact that former Figures 8 and 9 (now Figures 7 and 8 in the revised manuscript) changed slightly. Unfortunately, an earlier version of the centroids and covariances was accidentally used for the case study run (plausibility check), i.e. not the ones presented in our manuscript (see Fig. 4) and used for training and evaluation. The small differences do not affect our overall results and conclusions. We apologize for this minor error.

Figure 7. Estimated HPR^DF\_k for different hydrometeor classes applying HMC\_P to SR observations shown in Fig. 6. The black line indicates the clutter free bottom, the indigo line the freezing level height (DPR), and the STH for convective (magenta) and for stratiform (gray) SR-based RT. The dashed lines represent the bright band top and bottom. The bright band peak is illustrated as dash-dotted line

Figure 8. Estimated HPR^DP\_k for different hydrometeor classes with HMC\_P with GR observations shown in Fig. 6 The black lines indicate the Tt =  $4^{\circ}$ C and the indigo lines the Tb =  $0^{\circ}$ C isotherms. The Tc =  $-15^{\circ}$ C is indicated in gray/magenta for the GR-based stratiform/-convective RT.

Furthermore we updated the doi of the citation "A new aggregation and riming discrimination algorithm based on polarimetric weather radars" and we introduced the missing rain type RT in line 99.

Our response is highlighted in blue below. The revised manuscript with tracked changes is also provided for better transparency.

**Dear Authors,**

here are some comments I had to post to finalize to review process which took than due. Really sorry for this.

**General comments**

As clearly outlined in the Abstract, the objective of the study is threefold: to evaluate hydrometeor partition ratios retrievals, to exploit the combination of Dual Frequency (DF) satellite radar and Dual Polarization ground radar (GR) observations for estimating HPR based on satellite DF observations and to improve ground-based radar estimates of HPR.

NEXRAD S-band polarimetric radar observations are matched with satellite radar (SR) dual-frequency measurements of the Global Precipitation Measurement (GPM) satellite mission collected from 2014 to 2023.

The data, regardless of the platform (ground- or satellite-), undergoes a complex processing procedure to compensate for multiple error sources before being resampled into a common spatial domain.

The work is articulated in the following main steps:

- 1. Dual polarization ground-based radar data (NEXRAD) and dual-frequency Satellite-based radar data (GPM-DPR) are both resampled in so-called superobbed volumes to make the data resolution comparable. This aggregated volume contains hundreds of high-resolution radar gates.
- 2. The high-resolution radar gates contained in the superobbed volumes are classified with the Hydrometeor Classification algorithm proposed by Park et al. (2009) which assigns a dominant hydrometeor type (e.g. light rain, snow, hail, etc.).
- 3. For each aggregated volume  $S_i$ , it's determined the number of pixels  $N(HM_k)$  in which each hydrometeor class  $HM_k$  is dominant
- 4. For each class, the quasi-HPRs (qHPRs)are calculated as N(HMk) normalized by the total number of valid pixels in the superobbed volume.
- 5. These qHPRs are subsequently used both to train and to validate the HPR models on superobbed volumes from SR and GR observations. Consequently, the qHPRs represent a fundamental component of the study.

**Major comments**

This methodological choice raises a concern regarding potential circularity:

- The HPR estimates from GR are evaluated against qHPRs that are themselves derived from the same GR data (at higher resolution), which may lead to an overestimation of model performance.
- There is a risk that the HMCP model merely reproduces the classifications already present in the GR dataset, rather than demonstrating genuine improvement or generalization beyond the initial HMC algorithm.
- As a result, the **validation lacks independence**, which undermines the robustness of the conclusions regarding the model's effectiveness in the GR context.

To mitigate this issue, it would be advisable to use qHPRs **solely for training**, and to validate the GR-based HPR estimates using independent data sources (e.g., in situ observations, radiosonde profiles, or alternative/independent radar systems).

Thanks a lot for these critical comments and thus, pointing us to unclear and partly erratic formulations in the text. We assume that the following two erratic text segments of our manuscript, especially the last one, caused confusion:

- a) "qHPRs and averaged DF and DP variables serve as a basis for the HPR\_k^DF and HPR k^DP retrievals, which in turn are evaluated with the gHPRs."
  - -> We reformulated more precisely: "qHPRs and averaged dual-frequency and dual-polarization variables of the training dataset are used to derive covariances and centroids for each hydrometeor class. They serve as the basis for dual-frequency and dual-polarization based HPR retrievals within HMCP and are applied to the test dataset. The ensuing evaluation of HPR retrievals is performed with the qHPRs of the test dataset." (Line 11)
- b) "80% of the Si, including DF, DP measurements and qHPRs, serve as training data for the HMCP (Fig. 3 center) and the remaining 20% of the Si are utilized as training data set for the evaluation" —> This is for sure not true, instead the correct formulation is: "80% of the Si, including DF, DP measurements and qHPRs, serve as training data for the HMCP (Fig.3 center) and the remaining 20% of the Si are utilized as test data set for the evaluation". The first reviewer criticized this sentence as well. (Line 142)

Thus, we follow the conventional strategy to train an algorithm on a training data set and evaluate it on a separate test data set.

We also like to emphasize the challenges to provide a robust validation of HPRs, because airborne in-situ measurements with corresponding devices for a direct comparison in the entire precipitation column are expensive and thus rare. Additionally, the sample volumes of the in-situ devices represent only a small fraction of the actual radar volumes measured, which can lead to very large deviations in HPR measurements. In most cases, there is also a significant time lag

between airborne in-situ and ground-based radar measurements, and another problem arises from the definition of the various measured hydrometeor classes and the predefined ones of our HMC, just to mention a few challenges involved.

Point measurements from 2DVD video distrometers and to some extent also from laser distrometers may distinguish between hail, graupel, rain, and snow including the known uncertainties involved in these techniques , but are restricted to the surface and the coverage of these networks. Radiosondes may just provide the environmental information for the snow-rain transition

The validation with independent radar systems is very complicated. The comparison of HPR derived from NEXRADs S-band with other independent systems (X-band or C-band) would result in following challenges: a) Overlapping X/C-band data is more limited and to date not at our disposal. b) The entire procedure described in this paper would then have to be repeated for C-band or X-band, including the acquisition of a sufficiently large GR and SR data set (if available at all), c) The adjustment of the the scattering simulations to X-band and C-band for standard hydrometeor classification. d) The establishment of an adequate matching procedure between GR systems in order to be able to compare radar volumes of equal size.

As a consequence, we suggest and pursue an alternative strategy for validation exploiting quasi-HPRs estimated from higher-resolved radar measurements and compare those with our HPR retrievals based on the averaged radar variables. To avoid circularity, the covariances and centroides, which are exploited for the HMCP, are derived from the training dataset only. For the validation only the test data set is used which is independent of the training dataset. The calculated multivariate normal distributions are not influenced by the test data.

Additionally, we demonstrated the plausibility/consistency of the HMCP retrievals, confronting HMCP\_DF retrievals with HMCP\_DP retrievals for an independent case study with a GPM overpass over the KDDC NEXRAD ground radar (Sec. 4.3). The cross-comparison between the HPRs and the dual-frequency and dual-polarimetric variables also exploits the DP/DF gradients as proxies for microphysical processes, which are again reflected in the derived HPRs. This cross-comparison serves as an independent validation of the HPR-retrievals.

**Minor comments**

1. **Abstract**. In my opinion, there are too many acronyms in the abstract (about 40 in 15 lines), which makes it difficult to read. The abstract is the first thing people read. Please try to simplify it while ensuring it remains effective.

We fully agree and have significantly reduced the number of abbreviations in the abstract. It now reads as follows:

"Conventional radar-based hydrometeor classification algorithms identify the dominant hydrometeor type within a resolved radar volume, while newer techniques estimate the proportions of individual hydrometeor classes (hydrometeor partitioning ratios, HPRs) within a mixture. These newer algorithms (HMCDP) are based on dual-polarization

measurements from ground-based radars (GR), while to date no comparable algorithms for space-borne radars (SR) with dual-frequency capabilities exist.

This study (1) further improves HPR estimates based on GR dual-polarization measurements, (2) exploits the combination of dual-frequency SR and dual-polarization GR to introduce HPRs based on dual-frequency observations only, and (3) evaluates GR- and SR-based HPR retrievals.

To achieve these objectives, dual-polarization measurements of NEXRAD's GRs are matched with those of the dual-frequency precipitation radar of the Global Precipitation Measurement Core satellite. All matched volumes are represented by averaged dual-frequency and dual-polarization observations and several hundred GR sub-volumes classified with standard hydrometeor classification. The latter are used to calculate quasi-HPRs (qHPRs). qHPRs and averaged dual-frequency and dual-polarization variables of the training dataset are used to derive covariances and centroids for each hydrometeor class. They serve as the basis for dual-frequency and dual-polarization based HPR retrievals within HMCP and are applied to the test dataset. The ensuing evaluation of HPR retrievals is performed with the qHPRs of the test dataset.

HPRs show for most hydrometeor classes high correlations with the qHPRs and confirm the overall good HMCP performance. However, dual-polarization based classification performance is superior to dual-frequency ones. Both underestimate snow, overestimate graupel, and result in low correlations for big drops."

2. Introduction, pag. 1, line 18. I question the use of the adjective "fundamental". I'm not sure that HMC can really be considered fundamental to QPE at least at the range gate level. Do you mean from an operational or scientific point of view? Who actually applies HMC before estimating QPE? In stratiform precipitation, the application of VPR correction is much more important than any HRM classification. In convective precipitation, what is the overall impact of excluding certain hail spot from the rain field?

We assume you are referring to the adjective "essential". We claim that HMC plays an essential role to refine quantitative precipitation estimation, even though you are right that operational services do not apply HMC before estimating QPE. Instead, the German Weather Service e.g. is using different ZH thresholds and additional metrics to identify convective cores to optimize the choice of the rainfall retrieval, which at the end also serves e.g. as an estimate for hail occurence. In the scientific community, it is already shown that methods based on hydrometeor classification exhibit better performance compared to conventional algorithms. These advanced methods are adjusting the intercept parameters of the power-law relations depending on the classification results at each radar bin (hail, snow, rain, ...). Such methods were applied e.g. for NEXRADs WSR-88D in Giangrande and Ryzhkov (2008) (Eq. 7), Cifelli et al. (2011) or in Chen et al. (2017). We included the last two references in the revised manuscript. We are not

aware whether such methods are currently being used operationally, but from a scientific point of view prior HMC indeed enable to improve precipitation estimation.

3. To make the reading easier I suggest to focus the manuscript on the proposed methodology, moving the description of GR and SR data pre-processing (Sections n. 2.1, 2.2) to the Appendix.

Thanks for your suggestion. We agree to move the phase processing, attenuation correction, beam blockage calculations and calibration parts of the GR pre-processing to Appendix A (the standard HMC is now Appendix B), including also the figure for the pre-processing workflow to focus more on the HMCP methodology and results. However, we suggest leaving the temperature interpolation as well as the basic description of the GR data used in the main text, because they are important for the results and also help the reader to understand the ensuing text. We added in Line 115 "Additional GR processing e.g. phase processing, GR calibration and attenuation correction are explained in more detail in the appendix A."

Section 2.1 does not describe the SR pre-processing, instead this section informs the reader about the impact of different hydrometeor types on dual-frequency observations. Since no detailed HMC for DF measurements exist until now, we consider it helpful to keep Section 2.1 in the main text.